# Preparation of Progesterone Co-Crystals Based on Crystal Engineering Strategies

**DOI:** 10.3390/molecules24213936

**Published:** 2019-10-31

**Authors:** Huahui Zeng, Jing Xiong, Zhuang Zhao, Jingyi Qiao, Duanjie Xu, Mingsan Miao, Lan He, Xiangxiang Wu

**Affiliations:** 1Academy of Chinese Medicine, Henan University of Chinese Medicine, Zhengzhou 450046, Henan, China; hhzeng@hactcm.edu.cn (H.Z.); qiaojingyi618@126.com (J.Q.); xuduanjie123@163.com (D.X.); 2National Institutes for Food and Drug Control, Beijing 102629, China; xiongjing@nifdc.org.cn; 3Guangxi Institute for Food and Drug Control, Nanning 530021, Guangxi, China; keypersonal_1@sina.com

**Keywords:** crystal engineering, progesterone, co-crystals

## Abstract

Three co-formers of 2-chloro-4-nitroaniline (CNA), 2,5-dihydroxybenzoic acid (DHB), and 4,4′-biphenol (DOD) were selected to prepare the co-crystal of progesterone (PROG) based on crystal engineering strategies. These co-crystals were successfully obtained via slow evaporation from different solutions and were characterized by single-crystal X-ray diffraction spectroscopy, powder X-ray diffraction, IR spectroscopy, and differential scanning calorimetry. Different binding networks were observed in the co-crystal structures of PROG. The PROG-CNA co-crystal had the fastest rates and highest concentrations of PROG in PBS solution compared with PROG or other co-crystals in the dissolution experiments. This might be attributable to more stable and abundant interactions between the PROG and CNA molecules. Our investigations provide positive support for the selection of suitable co-formers using crystal engineering strategies.

## 1. Introduction

In recent years, novel crystal engineering strategies have been effectively developed to modify the characteristics of a variety of pharmaceutical products [1,2]. Among pharmaceutical solid-state complexes, co-crystals have attracted increasing attention. Co-crystals are molecular complexes composed of two or more constituents which are bound in the unit cell through various noncovalent interactions such as hydrogen bonding, π-π stacking, and Van der Waals interactions [3]. The preparation of co-crystals for a given substance offers the advantage of altering the physical and pharmaceutical properties of the drug without modifying its molecular formula. Specifically, co-crystals with the same active pharmaceutical ingredient (API) and different complementary co-crystal formers (CCFs) may show strikingly diverse characteristics, which are directly related to the nature of the CCF component, such as its melting point, solubility, bioavailability, hygroscopicity, and chemical stability [4,5]. Despite the importance of co-crystals, a systematic study of their structure and of the relation between the structure and the mechanical properties of the co-crystal still remains a rarity [6,7]. Consequently, the current development of pharmaceutical products relies, to a large extent, on empirical methods. Analysis and a deep understanding of the self-assembly rules and of the binding modes between the API and different CCFs may explain how the molecules interact with each other and how to select suitable CCFs to generate predictable structures. For this reason, crystal structure prediction constitutes the main focus of the vast majority of crystallographic articles.

Steroids play a prominent role in life sciences or as pharmacophores. Progesterone (PROG) is a naturally and poorly water-soluble steroid drug that belongs to the category of progestins, which are used in birth control pills and menopausal hormone replacement therapies [8]. The formation of co-crystals can improve properties of pharmaceuticals [9]. Karamertzanis has investigated the α···π stacking interaction of different steroids with aromatic molecules and has indicated that the steroid-binding affinity depends on the steroid backbone and especially the A-ring structure [10] (Figure 1). However, the solid-state complexation of PROG and aromatic molecules has never been systematically studied. In this work, PROG was selected as a co-crystallizing agent, not only due to its attractive backbone but also its terminal carbonyl groups, which provide the possibility for a more abundant network structure. Furthermore, PROG exists in two monotropically-related polymorphs: a thermodynamically stable form I, with a melting point at 129 °C, and a metastable form II, with a melting point at 122 °C [11,12]. The co-crystallization of PROG into a suitable co-former may contribute to improving its solid-state properties. However, co-crystallization may be influenced by many factors, including geometry, functional group position, and steric hindrance [13]. Fortunately, the interplay of several interactions in the co-crystal structure provides some guidelines upon which to choose its co-formers based on crystal engineering principles.

Although several PROG co-crystals have been reported in past years, a detailed and reliable structure-activity correlation is still yet to be established between the co-crystal structure and its mechanical properties [10]. As part of an ongoing study in crystal engineering, we investigated a series of co-crystals formed by PROG and 2-chloro-4-nitroaniline (CNA), PROG and 2,5-dihydroxybenzoic acid (DHB), and PROG and 4,4′-biphenol (DOD), see in Scheme 1. The co-former molecules were chosen based on their ability to interact with the carbonyl group of PROG via the formation of multiple strong hydrogen bonds. In this way, the carbonyl group of PROG acts as a hydrogen bonding acceptor while the co-former becomes a hydrogen bonding donor. This may promote the formation of co-crystals with an abundant weak interaction network. The preparation of the co-crystals was carried out using a slow evaporation technique. Several techniques were used to investigate the formation of the co-crystals, such as X-ray diffraction, thermal techniques, and spectroscopic techniques [14]. Single crystals of PROG-CNA, PROG-DHB, and PROG-DOD were obtained and characterized by X-ray diffraction to determine their crystal structures. This extensive investigation involved powder X-ray diffraction (PXRD), FT-IR spectroscopy, and differential scanning calorimetry (DSC) to validate the formation of co-crystals and their aforementioned binding modes. After the co-crystal screening, in vitro dissolution experiments were performed to explore the change in solubility and dissolution rates compared to bare PROG.

## 2. Results and Discussion

### 2.1. Preparation of Co-Crystals

The co-crystals were prepared using a slow evaporation technique. The same procedure was used for each co-crystal experiment; the only difference was in the volume of solvent and amounts of starting materials used. The general procedure involved dissolving equimolar amounts of PROG and the selected co-former in a suitable solvent and subsequently stirring at room temperature. After all compounds had been dissolved in solvent, the resulting solutions were filtered into vials and left for several days after each vial had been covered with Parafilm™. Various single crystals of the co-crystals were filtered for further screening.

### 2.2. Crystal Structure Analysis

As observed in the three co-crystals, four typical hetero-synthons (Scheme 2) were formed due to the strong hydrogen bonding network which involves the amino group, the carboxylic acid, and the hydroxyl group as donors and the carbonyl group (cycloketo and acetyl) as acceptors. These hydrogen bonding interactions strengthen the stability of the three co-crystals.

#### 2.2.1. Structure of the PROG-CNA Co-Crystal

The single crystal of PROG-CNA was obtained by the evaporation method from a mixed solvent of EtOH/water (1:1, *v*/*v*) containing a 1:1 molar ratio of PROG and CNA. X-ray diffraction analysis shows the expected crystal structure of PROG-CNA in 1:1 molecular stoichiometry was successfully obtained (Figure 2a). The final residual factor, *R*_1_ = 0.0375, was acceptable. Moreover, the PROG-CNA co-crystal belongs to the orthorhombic system and has an achiral space group of *P*2_1_2_1_2_1_, due to the chiral structure of PROG.

The CNA molecule contains a nitro group as a hydrogen-bonding acceptor and an amino group as a hydrogen-bonding donor. Those substituent groups transform CNA into a sort of “traffic hub” that holds together the interaction network from the different parts of the co-crystal. The two components are stabilized by the C-H···π and the C-H···O hydrogen-bonding interactions over a distance of 3.797 and 3.767 Å (Table 1 and Table 2), respectively. Head-to-tail adjacent PROG molecules are linked to a CNA molecule via synthons I and II (Scheme 2) with N_1_(B)···O_1_ and N_1_(B)···O_2_(C) distances of 2.952 and 3.031 Å, respectively. This induces the formation of a zigzag chain along the *c* axis (Figure 2c). In this case, the N atom of the CNA amino group adopts a *sp*^3^ hybridization with an H-N-H angle of 108.5°, which results in the formation of the hydrogen bonds with the carbonyl group of the PROG molecule. This value is close to the ideal hybridization angle of 109.5°. Moreover, the nitro group and the benzene ring of CNA are almost co-planar and they present a small dihedral angle of 6.594°. This angular deviation from an ideal co-planar situation decreases with the degree of conjugation between the nitro group and its benzene ring. This may be caused by the formation of the C-H···O hydrogen bond between the CNA homo-molecules and the hetero-molecules of CNA and PROG. Such a chemical conjugation enhances the stability of the 3D structure of the co-crystal, which is consistent with its calculated solid structure. Moreover, an $R_{3}^{3}\left(12\right)$ ring motif formed by three CNA molecules is observed in the structure of the PROG-CNA molecules which is established via a triple C-H···O hydrogen bond (Figure 2b). In addition, another $R_{3}^{2}\left(12\right)$ ring motif, multiple C-H···O/N hydrogen bonds (red dashed lines in Figure 2c) between PROG and the neighboring CNA molecules (Table 2), enhances the stability of the 3D structure of PROG-CNA. The overall stability of this co-crystal is strengthened by the α···π interactions occurring between the hetero-molecules of CNA and PROG. Hirshfeld surface analysis also shows the notable interactions, including the α···π, π-π, and weak C-H···O interactions [15] (Figure 2d). Interestingly, CDA (6-chloro-2,4-dinitroaniline), which is a molecule similar to CNA and contains one extra nitro group, forms vanillin isomer co-crystals. These structures are believed to form 1D-zipper-type tape upon dentate meshing and to generate a 2D-layer-type structure. Subsequently, these layers stack in an antiparallel fashion via weak π-π interactions that eventually build a layer-stack-type 3D structure. The 3D structure formation process of PROG-CNA is instead quite different and mostly influenced by the non-planar structure of the PROG backbone. Initially, a T-shape unit is located between two nearly vertical PROG molecules, which arrange themselves in a tail-to-tail fashion, while the structure assumes a sandwich conformation and an intermediate layer characterized by a linear shape-arrangement of coupled CNA molecules.

#### 2.2.2. Structure of the PROG-DHB Co-Crystal

DHB possesses the dual function of hydrogen-bond donors (carboxyl groups) and hydrogen-bond acceptors (hydroxyl groups) in the crystal. This property makes DHB an effective CCF in many co-crystals. Liao and co-workers have performed an exhaustive study of the position isomerism of the hydroxyl groups in a DHB molecule co-crystallization with piracetam [16]. Their investigations showed that the hydrogen bonding pattern of the co-crystals is quite different from that found in our works, despite the DHB isomers possessing the same kind and number of hydroxyl groups. Stilinović and co-workers have investigated the proton transfer phenomenon in a series of salts/co-crystals between DHB and its pyridine derivatives (CCFs) [17]. Their study showed that the value of Δp*K*_a_ has a significant influence on the solid formation of salts or co-crystals between DHB and CCFs. This empirical rule, based on the difference of p*K*_a_ values between a proton acceptor and a donor (Δp*K*_a_ = p*K*_a_ (base) − p*K*_a_(acid)), has been adopted by numerous scientists to predict the proton transfer effect. Furthermore, the molar ratio of the parent compound and the CCF may have a significant influence on the geometry of the co-crystal structure [1]. In this study, it is shown that DHB, which presents only one extra hydroxyl group in comparison with 4-HBA, is able to form a co-crystal with PROG as well.

A PROG-DHB crystal specimen was easily obtained by employing a method similar to the one used for PROG-CNA, with a solution of MeOH/water/ammonia (2:2:1, *v*/*v*/*v*) instead of EtOH/water (1:1, *v*/*v*) used in this case. X-ray diffraction analysis of PROG-DHB clearly confirmed a 1:1 molecular stoichiometry of PROG and DHB in the single crystal structure. The refinement of the PROG-DHB structure converged at an acceptable final residual factor *R*_1_ = 0.0502. This experiment shows that PROG-DHB adopts the same *P*2_1_2_1_2_1_ space group as PROG-CNA. This evidence may be correlated to the similar configuration of DHB and CNA.

The dual function of having hydrogen-bond donors and acceptors transforms DHB into an even more efficient “traffic hub” than CNA. In the PROG-DHB asymmetric unit, PROG and DHB are combined via the hetero-synthon III, designated an $R_{2}^{2}\left(8\right)$ ring motif, to form a hetero-molecule dimer (Figure 3a). This network involves several hydrogen bonds: (a) there is a strong hydrogen bonding interaction between the hydroxyl group of DHB and the acetyl carbonyl group of PROG with an O_5_···O_2_ distance of 2.620; (b) there is a weak C-H···O hydrogen bond between the methyl group and the carboxylic acid, with a C_21_···O_4_ distance of 3.643; (c) the PROG molecule links another DHB molecule through the cycloketo carbonyl group via a strong hydrogen bond with a O_6_(D)···O_1_ distance of 2.774 Å, and (d) the DHB molecules themselves are connected together through a series of C-H···O hydrogen bonds (Table 2), which generates a closed quadrangle cavity of about 7.1 × 16.7 Å^2^ dimensions arranged in an $R_{6}^{6}\left(46\right)$ ring motif (Figure 3b). The closed rings are connected in an orderly fashion via the DHB molecule and this drives the PROG-DHB to adopt a 3D geometry structure (Figure 3d). The DHB benzene ring is almost planar and its angular mean deviation is 0.0116 Å. Moreover, the carboxyl group is also arranged in a planar configuration with a dihedral angle relative to the benzene ring plane of 0.520°. This shows that the carboxyl group and the benzene ring are perfectly conjugated with each other. The remaining hydroxyl groups adjacent to the carbonyl group form an intramolecular hydrogen bond with an O···O distance of 2.612 Å. In addition, DHB acts as a hydrogen bond acceptor and it connects to the adjacent PROG molecular layer via a C-H···O interaction (Figure 3d). Further C-H···π interactions are also observed between the two components in the PROG-DHB co-crystal and in particular between the carbon atom and the centroid of the six-member ring (Cg) of DHB. These interactions, with a distance of 3.824 and 3.975 Å (Figure 3c and Table 1), further stabilize the solid structure of the co-crystal. As in the case of PROG-CNA, the α···π interactions strengthen the stability of the PROG-DHB co-crystal.

#### 2.2.3. Structure of the PROG-DOD Co-Crystal

A PROG-DOD crystal specimen was obtained by employing a procedure similar to the methods used for PROG-CNA and PROG-DHB. X-ray diffraction analysis shows that PROG-DOD is composed of PROG and DOD in a 1:1 molecular stoichiometry ratio. The refinement of the PROG-DOD structure converged at a final residual factor *R*_1_ = 0.0616. PROG-DOD was found to be a monoclinic crystal system and to have a chiral space group of *P*2_1_. This evidence constitutes a difference when comparing PROG-DOD with the PROG-DHB and PROG-CNA systems. This difference may be caused by the longer backbone of the DOD molecule.

As in the case of the DHB molecule, DOD has two hydroxyl groups which are positioned at both ends of the DOD molecule. Such a linearly-shaped molecular structure generates an extremely different interaction with PROG. In this case, DOD acts mostly as a “canal” that links the PROG molecules located at both ends of the structure rather than a “traffic hub” that holds together a complicated network. The PROG-DOD unit-cell volume measures 4121.1 Å^3^. This number is almost twice as large as that for CNA and DHB. This indicates that the three molecular pairs are arranged in an asymmetric unit (Figure 4a). The dihedral angles of the intramolecular benzene rings in the three DOD molecules are 8.291°, 14.449°, and 27.940°, respectively. For this reason, each single DOD molecule shows a different symmetry and the co-crystal structure belongs to a low symmetry space group. The PROG and DOD molecules form the hetero-molecule dimer via synthon IV and several interactions such as the O_3_A-H_3_A···O_1_ hydrogen bond over a distance of 2.747 Å and two C-H···O interactions with a distance of 3.634 Å (C_4_···O_3_A) and 3.438 Å (C_23_A···O_1_), respectively. The C_2_B-H_2_BB bond in the above PROG molecule interacts with its horizontally adjacent DOD molecule by forming a C-H···O bond. Moreover, a similar interaction exists between the DOD molecule and the adjacent PROG molecule located at the bottom of the structure via the C_2_B-H_2_BB···O_1_A and C_2_B-H_2_BB···O_3_B over a distance of 3.703 and 3.498 Å (C···O), respectively. This network generates a triangle-like motif (Table 3). The PROG-DOD dimers align along the [101] direction and partial dimers are further linked together to produce an extended ladder network in the (010) plane. The distance between two parallel dimers is about 6.0 Å. The 3D structure of the PROG-DOD molecule is generated by stacked ladders linked together via multiple C-H···O interactions. In this case, no C-H···π or α···π interactions were observed.

The DOD molecule connects to the PROG molecule in the co-crystal structure in a way which is similar to the 4-HBA-PROG. In this case, the stoichiometric ratio of the molecular structure is 2:1 and the homo-molecular dimers of 4-HBA self-assemble via $R_{2}^{2}\left(8\right)$ strong hydrogen bonds, which results in a linear co-crystal unit [8]. Similarly, Weyna and co-workers have reported that the DOD molecule co-crystallizes with linear bipyridyl compounds and adopts a zipper-type structure [18]. These examples show that a linear CCF can enhance the possibilities to predict the structure of the co-crystals.

### 2.3. IR Spectroscopy

The formation of novel polycrystalline phases obtained by employing a slow-evaporation technique was further confirmed by IR spectroscopy. The co-crystals show a characteristic spectral pattern which is similar to the ones of PROG and the CCFs. However, the frequencies of the C=O stretching vibration in the co-crystals appear blue-shifted due to the formation of the -C=O···H interaction between the PROG molecule and the CCFs. The IR spectrum of PROG-CNA shows (Figure 5) that in fact the two carbonyl C=O vibrations appear at 1627.91 cm^−1^ and 1585.48 cm^−1^, respectively, while in the PROG spectrum they are located at 1674.20 cm^−1^ and at 1612.49 cm^−1^. It is also possible to compare the positions of the C=O vibrations of the PROG-DHB and PROG-DOD co-crystals: in the former the peaks appear at 1658.77 cm^−1^ and 1612.48 cm^−1^ while in the latter at 1604 cm^−1^. The stretching frequencies of the C-H bonds of PROG in the co-crystals appear red-shifted when compared to the spectrum of the bare molecule. The peaks appear at 3371 cm^−1^, 3209 cm^−1^, and 3440 cm^−1^ in PROG-CNA, PROG-DHB, and PROG-DOD respectively. All these frequencies are clearly higher than 2943 cm^−1^, the frequency at which the C-H of the PROG molecule is located. This effect may be explained by the presence of a weak C-H···O or C-H···π interaction which induces the formation of more stable structures.

### 2.4. DSC and PXRD Analysis

The co-crystals were also characterized using PXRD and DSC to identify the formation of the co-crystal phase (Figure 6 and Figure 7). The PXRD experiments show new diffraction patterns in the co-crystals compared with PROG and the bare CCFs, and coincide with the simulated PXRD patterns, indicating the formation of a new co-crystal phase. The DSC profile of PROG-CNA presents an endothermic peak at 118.2 °C (∆H = −64.3 J/g) due to the melting point of the co-crystal. In the case of PROG and CNA this peak is visible at 131.5 °C and at 108.8 °C, respectively. The characteristic 2θ peaks of the PROG-DHB co-crystal are located at 9.79, 13.58, 14.60, 17.37, 17.77, 19.15, 20.25, 21.98, 25.39, and 27.62 degrees. This PXRD pattern shows the formation of the PROG-DHB co-crystal. The DSC profile of PROG-DHB presents an endothermic peak at 178.2 °C (∆H = −79.8 J/g) which appears at 131.5 °C in the case of PROG and at 212.3 °C in the case of DHB. The characteristic 2θ peaks of the PROG-DOD co-crystal obtained by PXRD are located at 7.85, 15.35, 16.81, 17.19, 17.52, 18.39, 20.95, 35.70, and 44.01 degrees. The DSC profile of PROG-DOD shows an endothermic peak at 199.5 °C (∆H = −42.2 J/g). In this case, the peak is also located at a different temperature compared to its bare constituents (PROG 131.5 °C and DOD 287.7 °C). The endothermic peak of the co-crystals are between API and CCFs, this shift in the position of the endothermic peaks is consistent with the majority of co-crystals.

### 2.5. In Vitro Dissolution

The results of the dissolution tests are overlaid in Figure 8. All the co-crystal forms of PROG demonstrate some improvement in solubility and dissolution rate compared to untreated PROG. Among the three CCFs, only PROG-CNA and PROG-DOD have an obviously faster solubility rate in the first 1 h, and then the maximum concentrations are only slightly increased compared with PROG. However, PROG-CNA shows a significant increase and a faster rate. The maximum concentrations measured for PROG-CNA are almost double their PROG counterparts. In addition, the HPLC chromatograms show that the samples have maintained the same retain time peak as PROG before and after the dissolution study (Figure 9). This may have a considerable impact on the bioavailability of PROG.

## 3. Materials and Methods

### 3.1. Materials and Reagents

All reagents and solvents used in these experiments are analytical reagent grade (A.R.) and were used without further purification unless stated otherwise. 2-chloro-4-nitroaniline (CNA, 98%, A.R.) and 2,5-dihydroxybenzoic acid (DHB, 99%, A.R.) were purchased from Ningbo Aike Biotechnology Co., Ltd.(Aike Biotechnology Co., Ltd., Ningbo, China), while 4,4′-Biphenol (DOD, 99%, A.R.) and progesterone (PROG, 99%, A.R.) were purchased from Heowns Biochem Technologies LLC. (Heowns Biochem Technologies LLC., Tianjin, China).

### 3.2. Synthesis and Crystallization

PROG-CNA: Progesterone (314.5 mg, 1.0 mmol) and an equal stoichiometric quantity of 2-chloro-4-nitroaniline (172.6 mg, 1.0 mmol) were placed in a clear and dry glass vial. The solvent EtOH/water (1:1, *v*/*v*) was added dropwise and the solution was mixed until the solid compounds dissolved completely. The glass vial was closed by using a serum cap with pinholes to allow the slow evaporation of the solvent. After 7 days, colorless block-like crystals could be observed. A sample of dimensions 0.05 × 0.04 × 0.03 mm was selected upon which to perform a single-crystal X-ray diffraction (SC-XRD) analysis.

PROG-DHB: The same preparation method for PROG-CNA was used for PROG-DHB. In this case, 2,5-dihydroxybenzoic acid was used (211.2 mg, 1.0 mmol) together with a MeOH/water/ammonia (2:2:1, *v*/*v*/*v*) solvent. Colorless block-shaped crystals were produced upon solvent evaporation. A specimen of dimensions 0.06 × 0.05 × 0.04 mm was selected upon which to conduct a SC-XRD experiment.

PROG-DOD: The preparation of PROG-DOD was similar to the PROG-CNA preparation. In this case, 4,4′-biphenol (186.2 mg, 1.0 mmol) was used instead of CNA. Upon the evaporation of the EtOH/water (1:1, *v*/*v*) solvent, colorless block-shaped crystals were collected. Moreover, a sample of dimensions 0.07 × 0.06 × 0.05 mm was selected for an SC-XRD experiment.

### 3.3. Differential Scanning Calorimetry

DSC curves were measured with a Mettler Toledo DSC 822^e^ calorimeter in an aluminum sample pan with nitrogen atmosphere. The flow rate was kept at 50.0 mL/min during the experiment and the constant heating rate at 10 °C/min.

### 3.4. IR Spectroscopy

Fourier transform infrared spectroscopy was performed using a PerkinElmer Spectrum 100 Spectrometer (PerkinElmer, Waltham, MA, USA, country) with KBr pellet at room temperature. The spectrum of a sample was collected in the 4,000–450 cm^−1^ range.

### 3.5. Powder X-Ray Diffraction

PXRD measurements were carried out with a Rigaku Smartlab 9kw diffractometer with a radiation wavelength of 1.5418 Å generated by a Cu tube. The spectrometer was operated at 45kV and 200 mA and continuously scanned from 5° to 50°. The samples were powdered and placed randomly on a silicon surface in order to avoid any preferred orientation.

### 3.6. In Vitro Dissolution

The tests were carried out on an RC806D dissolution apparatus (Tianjin Tianda Tianfa Technology Co., Ltd., Tianjin, China) with the United States Pharmacopeia (USP) apparatus II (paddle-type). The dissolution media was purified water. Before the dissolution study, the powders were sieved using a 200 mesh screen, a step which was repeated three times. Progesterone (30 mg) and progesterone co-crystals corresponding to 30 mg of progesterone were added to 100 mL of dissolution media. The rotation speed was set at 100 rpm with a dissolution bath temperature of 37.0 ± 0.5 °C. Aliquots of the dissolution medium were withdrawn at 5, 10, 15, 20, 30, 40, 60, 90, 120, and 180 min. The dissolution media was replaced with an equivalent volume of fresh media at 37 °C. The withdrawn slurry was filtered and progesterone concentrations were determined using HPLC. This method used a chromatographic system with Empower 3 software (Waters Technology, Milford, MA, USA) which was equipped with a photo-diode array (PDA) detector. The mobile phase consisted of methanol, acetonitrile, and water (25:35:40). The mobile phase was degassed and pumped at a flow rate of 1.0 mL/min through a reversed phase column 250 mm × 4.6 mm (i.d.) BDS Hypersil C8 with an average particle size of 5 µm. The effluent was detected at 241 nm.

### 3.7. X-Ray Crystallography

Crystal data and structure refinement details are summarized in Table 4. All the hydrogen atoms bound to the carbon atoms were positioned in predefined positions which have been previously calculated. These positions were then refined using a riding model with C-H = 0.93–0.99 Å, *U*_iso_(H) = 1.2*U*_eq_(C), and *U*_iso_(H-Methyl) = 1.5*U*_eq_(C-Methyl). All the active hydrogen atoms (OH and NH) were found in difference Fourier maps. During the final refinement cycles, OHs and NHs were repositioned in the initially calculated geometrical coordinates. These positions were then refined using the same riding model, with O-H = 0.82 Å, N-H = 0.87 Å, *U*_iso_(H) = 1.2*U*_eq_(N), and *U*_iso_(H) = 1.5*U*_eq_(O).

## 4. Conclusions

In this work, co-crystals of progesterone with 2-chloro-4-nitroaniline, 2,5-dihydroxybenzoic acid, and 4,4′-biphenol were synthesized by slow evaporation and with a molar ratio of 1:1. The crystal structures have been presented and discussed in detail in this paper. Specifically, CNA and DHB, which present similar molecule configurations, show a rather diverse hydrogen bonding network when they co-crystallize with PROG. Both CNA and DHB work as a powerful “traffic hub” which generates a complex hydrogen bonding pattern in the co-crystal structure and multiple weak interactions. The linear shape of the DOD molecule generates a broader connectivity pattern that, to some extent, may help in the prediction of the crystal/co-crystal structure. Additionally, all the co-crystal structures were confirmed by powder X-ray diffraction spectroscopy, IR spectroscopy, and differential scanning calorimetry. The dissolution tests indicate that faster rates and higher concentrations of PROG were attained after the co-crystallization, especially for PROG-CNA. This might be attributable to more stable and abundant interactions which are composed of multiple C-H···N hydrogen bonds and α···π interactions between CNA and PROG. In the long term, the investigation of crystal structures provides positive support for the selection of suitable co-formers using the crystal engineering strategies.

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
