# Peer review of "Preparation of Progesterone Co-Crystals Based on Crystal Engineering Strategies"

_molecules, 2019, doi:10.3390/molecules24213936_

Round 1

Reviewer 1 Report

This is a nice, rigorous study of three co-crystals of progesterone. The inclusion of dissolution data is particularly helpful. The work is well done and I am happy to recommend publication. I wonder whether the authors tried other co-formers? It would be surprising if they only tried three and all three worked. It would be helpful to also report the co-formers that did NOT form co-crystals to make this a more consistent and complete study. I note one typo "[Error! Bookmark not defined.]". I am not convinced by the long and geometrically strange CH...O2N interactions. Have the authors considered using the Hirshfeld surface technique in Crystal Explorer to make a more holistic analysis of the crystal packing?

Author Response

We have tried many other amino-acid co-formers in this study, such as histidine, lysine, arginine, threonine, isoleucine, and even saccharin, but these did not work. All typos in including "[Error, Bookmark not defined]" were revised in the original manuscript. Hirshfeld surface analysis also shows the notable interactions,  including the α···π, π-π interactions,and weak C−H···O interactions. (Figure 2. (d))

Reviewer 2 Report

The manuscript "Preparation of Progesterone Co-crystals Based on Crystal Engineering Strategies" describes the cocrystallization behavior of a steroid Progesterone (PROG) with three coformers 2-chloro-4-nitroaniline (CNA), 2,5-dihydroxybenzoic acid (DHB) and 4,4'-biphenol (DOD). Authors have clearly demonstrated the crystal structures, dissolution methods, and other analytical methods.

Page 1/line 14: these are not novel coformers .. change the wording

Page2/line 48: takeout 'always' as there are few examples that cocrystals can't show better properties than the parent compound

Page 2/line 58: Insert reference for this

Page 2/line 62: Check the error

Page 2/line 78: it is a figure as there is no scheme here, please change the description accordingly.

page 7/line 227: R in ring representation shouldn't be italics

page 7/line 228: Check the error

page 7/line 234: If possible please insert the hydrogen bond interactions for Progesterone reported structure as well.

page 10/line 263: Figure 5: Please include the three figures of simulated PXRD pattern (from cif file) with the powder XRD pattern (stacked version), so the purity of bulk sample that has been used for other studies can be understood. Instead of a physical mixture, you can compare a. coformer. b. cocrystal and c. progesterone, so it could be easy for a reader.

page 10/line 266: Figure 6: if possible please insert the endothermic values and heat of enthalpy values in the figure. (correct the figure description)

page 11/line 283: Here it seems you have done powder dissolution, may I know what is the micron size of the powder you have used. As you know the powder should be sieved before the powder dissolution study be conducted to get the uniform results. Also, how many times have you repeated this experiment?

Also, How do you know the samples are stable even after the dissolution study? I would suggest you, to keep the HPLC chromatograms before and after the dissolution study of the compounds in the manuscript.

page 13/ line 353: Please insert the Goodness of fit (Goof) factor in the crystallography table.

Also, please rewrite the conclusion (shorten) and correct the grammar. Also, are the coformers belongs to GRAS category? if not I think no point in conducting in vivo studies.

"Coformer" is written as "conformer" throughout the manuscript please correct this accordingly.  

Author Response

Page 1/line 14: these are not novel coformers ... change the wording

Reply: The word “novel” in original manuscript was deleted.

Page2/line 48: takeout 'always' as there are few examples that cocrystals can't show better properties than the parent compound

Reply: The word “always” in original manuscript was deleted.

Page 2/line 58: Insert reference for this

Reply: Stahly, G. P. Diversity in Single- and Multiple-Component Crystals. The Search for and Prevalence of Polymorphs and Cocrystals. Crystal Growth and Design 7(6):1007-1026

Page 2/line 62: Check the error

Reply: This is a typo; we have revised in original manuscript.

Page 2/line 78: it is a figure as there is no scheme here, please change the description accordingly.

Reply: Scheme 1. Chemical structures of the progesterone (PROG) and the co-formers of 2-chloro-4-nitroaniline (CNA), 2,5-dihydroxybenzoic acid (DHB), and 4,4'-biphenol (DOD).

Page 7/line 227: R in ring representation shouldn't be italics

Reply: We completed this modification of italics.

Page 7/line 228: Check the error

Reply: This is a typo; we have revised it in original manuscript.

page 7/line 234: If possible please insert the hydrogen bond interactions for Progesterone reported structure as well.

Reply: Figure 1.The crystal structure of progesterone∙ 9-phenanthrol  complex displaying an α⋯π dimer and a hydrogen-bonded neighboring PROG molecule (blue).

Page 10/line 263: Figure 5: Please include the three figures of simulated PXRD pattern (from cif file) with the powder XRD pattern (stacked version), so the purity of bulk sample that has been used for other studies can be understood. Instead of a physical mixture, you can compare a. coformer. b. cocrystal and c. progesterone, so it could be easy for a reader.

Reply: I have replaced the original XRD figures according your advice.

Page 11/line 283: Here it seems you have done powder dissolution, may I know what is the micron size of the powder you have used. As you know the powder should be sieved before the powder dissolution study be conducted to get the uniform results. Also, how many times have you repeated this experiment? 

Reply:  Before the dissolution study, the powders were sieved by using the 200 mesh screen, and repeated three times.

Also, How do you know the samples are stable even after the dissolution study? I would suggest you, to keep the HPLC chromatograms before and after the dissolution study of the compounds in the manuscript.

Reply: I have kept the HPLC chromatograms before and after the dissolution study, and the samples have contained the same retain time peak with PROG during the dissolution study as follows:

Figure 9.The HPLC chromatograms of the PROG(3 times) and the PROG-CNA, PROG-DHB and PROG-DOD after 180 min in the dissolution study.

page 13/ line 353: Please insert the Goodness of fit (Goof) factor in the crystallography table.

Reply: I have relabeled the Goof value in the crystallography table.

Also, please rewrite the conclusion (shorten) and correct the grammar. Also, are the conformers belongs to GRAS category? if not I think no point in conducting in vivo studies.

Reply: Don’t belong to GRAS category, and I have deleted the in vivo studies conducting in the manuscript.

"Coformer" is written as "conformer" throughout the manuscript please correct this accordingly.  

Reply: We completed these modifications in original manuscript.
